# Choline Metabolites, Hydroxybutyrate and HDL after Dietary Fiber Supplementation in Overweight/Obese Hypertensive Women: A Metabolomic Study

**DOI:** 10.3390/nu13051437

**Published:** 2021-04-24

**Authors:** Carla Patricia Novaes dos Santos Fechine, Mussara Gomes Cavalcanti Alves Monteiro, Josean Fechine Tavares, Augusto Lopes Souto, Rafaella Cristhine Pordeus Luna, Cássia Surama Oliveira da Silva, Jairo Alves da Silva, Sócrates Golzio dos Santos, Maria José de Carvalho Costa, Darlene Camati Persuhn

**Affiliations:** 1Postgraduate Program in Nutrition Sciences, Federal University of Paraíba, João Pessoa 58059-900, Brazil; mussara.monteiro@hotmail.com (M.G.C.A.M.); rafaellacristhine@yahoo.com.br (R.C.P.L.); suramajpa2@hotmail.com (C.S.O.d.S.); jairoalvess28@gmail.com (J.A.d.S.); mjc.costa@terra.com.br (M.J.d.C.C.); darlenecp@hotmail.com (D.C.P.); 2Department of Pharmaceutical Sciences, Federal University of Paraiba, João Pessoa 58059-900, Brazil; josean@ltf.ufpb.br (J.F.T.); augustosouto@gmail.com (A.L.S.); socratesgolzio@ltf.ufpb.br (S.G.d.S.)

**Keywords:** dietary fiber, hypertension, obesity, metabolomics, nuclear magnetic resonance

## Abstract

Metabolomics has been increasingly used to evaluate metabolic changes associated with morbidities. The objective of this study is to assess the metabolic profile before and after intervention with mixed dietary fiber in overweight and obese hypertensive women. This is an intervention study, and the sample consists of 14 women aged 28 to 58 years. An intervention with 12 g of mixed soluble and insoluble fiber is performed for a period of eight weeks. Serum metabolites are identified using a Bruker ^1^H NMR spectrometer at 400 MHz. Multivariate data analysis, including principal component analysis (PCA), is used to differentiate the two groups. After supplementation with dietary fiber, there is a significant increase in the peak intensity values of the metabolites HDL-C (0.0010*), choline (0.0012*) and hydroxybutyrate (0.0010*) as well as a decrease in systolic (0.0013*) and diastolic (0.0026*) blood pressure. The analysis of the metabolomic profile allows the identification of metabolites that have been associated in the literature with hypertension and excess weight (choline, hydroxybutyrate and amino acids) and with fiber intake (choline, hydroxybutyrate and amino acids) in addition to an increase in HDL-C. The increase in the detection of the described metabolites possibly occurs due to the presence of pathologies and the use of fiber in the intervention, which also contributes to elevated HDL-c and reduced blood pressure.

## 1. Introduction

Despite the scientific advances in treatment, identification of pathophysiological mechanisms and public policies implemented in recent decades, hypertension remains a public health problem due to its high prevalence worldwide [1]. Considered one of the main causes of morbidity and mortality, it is an important risk factor for cardiovascular diseases (CVDs) and renal complications [2,3,4,5]. The etiology of hypertension is heterogeneous and multifactorial. This condition occurs as a consequence of genetic, environmental and lifestyle factors that trigger biological factors such as oxidative stress, endothelial dysfunction and the functionality of the renin-angiotensin-aldosterone system and the sympathetic nervous system [6,7].

Metabolomics has been used to evaluate metabolic disorders associated with hypertension. Studies comparing the metabolic profile of hypertensive patients to healthy ones point to changes in the metabolites involved in the metabolism of amino acids and lipids as the most prominent factor [8,9,10,11]. With advancements in technology, a large number of metabolites can be measured in various body fluids, such as urine, feces, saliva and blood, providing more information on which metabolic pathways may be affected after exposure to several factors, especially food intake [9,12].

Considering the persistent increase in the prevalence of hypertension, the need for innovative prevention and control strategies has become important. The influence of fiber intake has been a topic discussed since the mid-1970s and this practice has already been associated with the prevention of cardiovascular disorders, with improvements in blood glucose, insulin resistance, weight loss and blood pressure (BP). However, there is still no consensus on its recommendation for the prevention and treatment of hypertension [13,14].

The increase in fiber intake by the general population seems to contribute to the prevention of hypertension, although there are controversies [15,16,17]. The decrease in BP values is due to soluble dietary fiber as well as to an increase in intestinal viscosity, delaying nutrient absorption and inhibiting the absorption of cholesterol and bile acids [18,19].

Current studies relating hypertension and metabolomic profiles are still in the initial stages. Therefore, more evidence and interpretation are needed, especially when related to cases with associated morbidities that are often found in the population, such as obesity and hypertension, as well as studies using interventions with specific nutrients. Thus, the present study aims to examine the metabolic profile of overweight and obese hypertensive women before and after intervention with mixed dietary fiber, a topic unreported in the literature.

## 2. Materials and Methods

### 2.1. Study Design and Sample Characteristics

We conducted a randomized intervention study in which 14 overweight/obese hypertensive women, aged 20 to 50 years, used fiber supplements (12 g/day of dietary fiber). Women with a medical diagnosis of hypertension and with high BP values were invited and recruited at the Blood Donor Centre in João Pessoa, Paraíba (PB), Brazil.

The initial number of participants was 20, however, there was sample loss due to the inclusion criteria (*n* = 5) and withdrawal due to the occurrence of adverse effects due to the use of fiber (colic and diarrhea *n* = 1). 

This study was submitted to the Research Ethics Committee of the Healh Sciences Center, Federal University of Paraíba and was approved under CAAE number 64573917.4.0000.5188 and registered in the Brazilian Registry of Clinical Trials under TRIAL: RBR-2PH4F9. The intervention with fiber was performed for a period of eight weeks; fiber was packed in sachets (12 g) and consisted of soluble fiber (7 g) and insoluble fiber. The fiber supplement was prepared in a pharmacy in João Pessoa, PB, Brazil. The composition of the supplement, which contained a nutraceutical based on mixed fibers, was soluble fiber (7 g), guar gum (4 g; lot ALL 0605354), NutraFlora (FOS; 1 g; lot Galena (CIQ): 1505006204), psyllium (2 g; scientific name *Plantago ovata*; manufacturer batch: 949/2013) and insoluble fiber (5 g; microcrystalline cellulose 101—5 g, batch 14116094A, manufacturing batch C1404014, formula C6nH10n + 205n = 1).

In this study, only women diagnosed with primary hypertension with a body mass index (BMI) of 25–35 m²/kg and BP above 130/80 mmHg were included.

The exclusion criteria were diabetes, liver failure, congestive heart failure (grades 3 and 4), renal failure with creatinine values greater than 3.0 mg/dL and secondary hypertension or systolic BP values ≥ 180 mmHg or diastolic BP values ≥ 110 mmHg.

The women who agreed to participate in the study kept their eating and physical activity habits stable for at least four weeks before beginning the intervention and reported not having the intention to change those behaviors during the study.

### 2.2. Analysis of Biochemical Parameters

Lipid profile analyses were performed before and after the intervention of serum cholesterol (TC), triglycerides (TG), high-density lipoproteins (HDL-C), low-density lipoproteins (LDL-C), fasting glucose and high-sensitivity C reactive protein (hs-CRP). For biochemical analyses, blood was collected by venipuncture in three different sterile tubes: tube 1 (with the anticoagulant K_3_ EDTA-ethylenediamine tetra-acetic acid), tube 2 (with the anticoagulant sodium fluoride) and tube 3 (with the clot activator). The samples from tubes 2 and 3 were centrifuged immediately to obtain plasma and serum, respectively, and were subjected to analysis less than 2 h after collection.

The lipid fractions, TC and TG were determined by enzymatic assays [20,21]. TC and TG (enzymatic method—Trinder) were measured in serum aliquots in an automated analyzer using a Labtest kit.

### 2.3. Blood Pressure Assessment

Blood pressure was assessed using the criteria proposed by the Brazilian Society of Cardiology in its Hypertension Guideline of 2010. The procedure consisted of performing three BP measurements, using an OMRON HEM-742INT blood pressure monitor, with a 1-min interval between each measurement. The women were resting (for at least 5 min) in the sitting position with their feet on the ground and arms supported on the table.

### 2.4. Nutritional Evaluation

To calculate BMI, the body weight (kg) was divided by height (meters) squared, adopting the cut-off values recommended by the World Health Organization (WHO) for adults aged 20 to 59 years: <18.5 kg/m^2^ (underweight), 18.5–24.9 kg/m^2^ (normal weight), 25.0–29.9 kg/m^2^ (overweight), 30.0–39.9 kg/m^2^ (obese), and ≥40 kg/m^2^ (extremely obese) [22]. Weight and height were measured in triplicate, taking the mean of the three values [23].

The usual intake of total calories, carbohydrates, proteins, lipids and fibers were analyzed using two recalls (24-h food recall), applied by professional nutritionists on the first day of the study and after eight weeks. Food consumption was analyzed using the Virtual Nutri Plus software and was subject to intra and interpersonal adjustment using the MSM software (multiple sources method).

The usual intake of total calories, carbohydrates, proteins, lipids and fibers was 2390.91 ± 1449.42 kcal, 267.82 ± 103.26 g, 68.29 ± 25.69 g, 61.06, ± 30.90 g and 15.57 ± 5.59 g, respectively, after analysis of two 24-h food recalls, applied on the first day of the study and at eight weeks. The usual intake of dietary fibers in the diet, before the intervention and after, was 15, 57 ± 5.59 g. That added to 12 g of fibers from the intervention with mixed dietary fibers, totaling 27, 57 g of mixed fiber consumption during the eight weeks of the study.

### 2.5. Metabolomic Profile

Metabolomic profile analyses were carried out using ^1^H nuclear magnetic resonance (NMR) at the Multi-User Laboratory of Characterization and Analysis (LMCA, for its initials in Portuguese) of the Federal University of Paraíba (UFPB, for its initials in Portuguese) using a Bruker spectrometer at 400 MHz. The analysis of the women’s serum was carried out pre- and postintervention. Initially, 5 mL of venous blood was collected in polypropylene tubes containing EDTA in the early morning under fasting conditions. After collection, the blood was centrifuged at 3000× *g* for 10 min at 4 °C to isolate the serum. The serum was transferred to a sterilized tube and stored at −80 °C; it was thawed for use, removing 300 µL of serum and adding 300 µL of phosphate buffer (TSP) as a reference standard, and homogenized by vortexing for 30 s, followed by centrifugation at 14,000× *g* for 10 min at 4 °C. After centrifugation, 60 µL of heavy water was added. This was followed by NMR analysis; the supernatant, in a 5 mm tube, was transferred to a 400 MHz NMR. The pulse sequence used to obtain data was CPMG-BRUKER and was performed in triplicate.

The obtained spectra were processed using MestreNova software (version 6.02, MestreLab Research S.L.-Spain), and the chemical shifts were expressed in PPM. The metabolites were identified by comparing the obtained chemical shifts with chemical shifts of authenticated samples available in the human metabolome database HMDB The spectra were referenced by tetramethylsilane (TSP), in a total range chosen for analysis of between 0.1 and 9 ppm, with *binnings* of 0.02 ppm, generating a data matrix of 96 samples and 225 variables that was exported in ASCII format to the statistical software “Unscrumbler” (version 9.7, CAMO Analytics AS, Norway). The data were not normalized by area, and the peaks for TMS, D_2_O, methanol and acetone were discarded. 

### 2.6. Statistical Analysis

To describe the general characteristics of the groups, data are expressed as the mean and standard deviation, proportion or N (%). Continuous variables were tested for normality and homogeneity of variance using the Shapiro-Wilk test. To evaluate possible pre- and postintervention differences in relation to the studied metabolites, paired t-tests or Wilcoxon tests were used based on the distribution of the variables. The Spearman correlation test was applied to determine the existence of a relationship between systolic and diastolic BP and the metabolites (choline, hydroxybutyrate, alpha-glucose and HDL). The statistical software R version 3.3.2 was used for analyses, and statistical significance was set as *p* < 0.05.

The NMR data were processed using MestreNova software (version 6.02, MestreLab Research S.L.). Principal component analysis (PCA) was used and the chemical shifts were expressed as PPM.

## 3. Results

The characteristics of the sample regarding demographic, epidemiological and lifestyle variables are presented in Table 1. 

The lipid profiles and BP values were compared before and after the intervention, and there was a difference between the HDL cholesterol and BP values (Table 2). Observing the lipid profile of the 14 women who participated in the present study, 35.7% had HDL-C values below those recommended, and 42.9% of the participants had high TG levels.

The usual food consumption of total calories, carbohydrates, proteins, lipids and fibers was, respectively, 2390.91 ± 1449.42 kcal, 267.82 ± 103.26 g, 68.29 ± 25.69 g, 61.06 ± 30.90 g and 15.57 ± 5.59 g.

Figure 1 and Figure 2 show, based on the figures, the most prominent metabolites before and after the intervention.

The following metabolites were notable based on the peak area (* = PPM—chemical shift): *0.88 (HDL); *0.84 (HDL); *1.32 (HDL); *1.28 (HDL); *3.25 (choline); *3.65 (choline); *1.24 (HDL); *1.20 (3-hydroxybutyrate); *4.78 (water); *4.90 (alpha-glucose); *4.94 (HDL).

The representative 1D Carr Purcell Meiboom Gill (CPMG) NMR spectrum of hypertensive patients after fiber supplementation is presented in Figure 3. It was possible to identify the following main metabolites: lipids, glucose, valine, leucine, isoleucine, alanine, choline, tyrosine, histidine and 3-hydroxybutyrate. Spectral area assignments were based on a 1H-NMR spectrum representative of serum. The assignment of NMR to metabolites was based mainly on literature research.

The metabolites that were significant for HDL, choline, and hydroxybutyrate when comparing the groups, as presented in Table 3. Regarding the Spearman correlation test, there was no correlation between metabolite values and BP after the intervention with mixed dietary fibers (results not shown).

## 4. Discussion

In the present study, we identified an increment in HDL-C, choline and hydroxybutyrate metabolites and reduction in systolic and diastolic BP in hypertensive women with excess weight, after dietary fiber supplementation. In the literature consulted, there were no studies related to the metabolomic profile of overweight and obese hypertensive women who consumed mixed dietary fiber.

Regarding metabolites studies conducted with women and intervention with fiber, an analysis was found of the serum of hypercholesterolemic postmenopausal women, who consumed a minimum of 20% of their daily energy intake as rye bread rich in insoluble fiber, with lower amounts of leucine and isoleucine and higher amounts of betaine [24,25]. 

Changes in these metabolites are often related to intestinal microbiota dysfunction. Branched-chain amino acids (valine, leucine and isoleucine) are associated with insulin resistance and obesity (a morbidity included in the sample of the present study), probably due to excess energy intake, as intestinal microbiota are an important modulator of branched-chain amino acid levels [26].

The study described above indirectly corroborates the increase in choline found in the present study because betaine is derived from choline, and this metabolite is also positively related to obesity, lipid metabolism and hypertension [25,27,28,29,30,31]. 

The differences in the results regarding the metabolites found are possibly due to the fiber composition, which in the present study was rich in soluble fiber and originated from various foods, such as guar gum, NutraFlora (FOS), psyllium seed bark and cellulose. The characteristics of the studied populations also differ, although both are composed of women; in the present study, obesity and hypertension were inclusion criteria. 

A study conducted in premenopausal overweight/obese women with (*n* = 36) and without metabolic syndrome (*n* = 42) found that branched-chain amino acids (leucine, isoleucine and valine), aromatic amino acids (phenylalanine, tyrosine, tryptophan and methionine) and several types of fatty acids and phospholipids were associated with metabolic syndrome [32].

The results [32] partially corroborate those of the present study, which also identified the hydroxybutyrate metabolite and an increase in phosphatidylcholine (rich in choline), as well as branched-chain amino acids, histidine, tyrosine, arginine and alanine, although they were not significantly different after the intervention with dietary fiber.

Bagheri et al. [33] used a metabolomic approach to differentiate specific metabolites of healthy and obese individuals of both sexes (18 to 50 years of age) to identify profiles of metabolites associated with obesity.

Those authors observed that some amino acids and polar lipids, such as branched-chain amino acids and phosphatidylcholine, were higher in obese individuals. Thus, certain choline-based metabolites, such as phosphatidylcholine and lysophosphatidylcholine, are commonly altered in the presence of obesity, indirectly corroborating the present study regarding the significant increase in choline in the sample studied.

Choline at adequate levels is an essential nutrient for maintaining health, including adequate lipid metabolism [30], and is involved in the pathogenesis of metabolic syndrome and cardiovascular diseases [34,35]. Gao et al. [30] observed that adequate choline and betaine intake was associated with a lower percentage of body fat and higher lean mass, specifically in men, corroborating the present study, which did not find an association of these metabolites with body composition in women. However, the mechanisms by which choline and betaine improve body composition are still unclear.

Different mechanisms have been proposed to explain the effects of short-chain fatty acids, such as butyrate, on the control of obesity [34]. Butyrate attenuates diet-induced obesity and insulin resistance in mice and negatively regulates PPAR-γ expression and activity, thus promoting a change in lipogenesis toward lipid oxidation, thus stimulating oxidative metabolism in the liver and adipose tissue and increasing energy expenditure [36,37,38,39]. In their recent study, Wang et al. [34] found a positive association between BMI and short-chain fatty acids (butyrate and isobutyrate) measured by plasma metabolomics in Chinese adults aged 30–68 years. In addition, butyrate has anti-inflammatory effects that are presumed to be mediated by the inhibition of histone deacetylase (HDAC).

Regarding studies with metabolomic profiles of hypertensive women, no studies with this objective were found in the literature. A recent study with a sample that included both sexes developed by Goïta et al. [29] stands out; they evaluated plasma metabolic profiles using metabolomics by means of mass spectrometry, in 64 individuals aged between 34 and 60 years, 28 of whom were hypertensive (13 women and 15 men) and 36 nonhypertensive (18 women and 18 men).

Goïta et al. [29] identified a considerable increase in the levels of phosphatidylcholine in the blood of hypertensive patients of both sexes. In women, other metabolites increased, including the amino acids arginine, leucine, isoleucine, tryptophan, phenylalanine, tyrosine, lysine, glutamine, histidine, methionine and citrulline. In part, the results of the present study corroborate this author by observing a significant increase in choline and identifying leucine, which may indirectly have contributed to the significant increase in hydroxybutyrate. Moreover, the studies are coincident in other amino acids such as arginine, isoleucine, tyrosine, glycine, valine, alanine and histidine, although without differences in the comparison between pre- and post-intervention samples. Choline is recognized as an essential nutrient for maintaining human health, as it is involved in the donation of a methyl group and the synthesis of phospholipids, lipoproteins and betaine and is a precursor of acetylcholine [30,40,41,42].

Regarding studies on the lipid profile of obese and hypertensive individuals with or without fiber intake, Pasanta et al. [43] investigated the lipid profiles of serum metabolites using 1H NMR in 46 young obese and healthy adults of both sexes (18 to 25 years of age). The spectrum for the overweight group showed slightly higher lipids, α-glucose and β-glucose. Therefore, the TG, LDL, VLDL, HbA1c and blood glucose levels were significantly higher in the obese group, in addition to significantly lower HDL. In the present study, the mean HDL-C values were within the reference values, and after the intervention with fiber, these values increased significantly (Table 2), subsequently leading to a significant increase in HDL metabolites.

In a study by Pasanta et al. [43], there was a nonsignificant increase in choline in the obese group compared to the healthy group, suggesting a change in lipid metabolism, thus corroborating the present study in which choline was the metabolite that stood out the most in women after fiber supplementation.

Thus, scholars have supported an association between lipid metabolism and BP [11]. Dietary fiber intake seems to promote adequate serum lipid levels, and this benefit may be associated with improved endothelial vasodilation and oxidative stress, and consequently, decreased BP levels [44,45].

We observed a significant increase in butyrate in the metabolites derived from the serum of women after eight weeks of mixed fiber intake. This is a short-chain fatty acid that is produced in large quantities by bacterial fermentation of dietary fiber in the intestinal lumen. Butyrate can regulate the secretion of apoA-IV and, therefore, modulate reverse cholesterol transport, which may explain the increase in HDL-C in the present study [46,47].

In addition to stimulating the secretion of lipoprotein particles containing apoA-IV, favoring the removal of cholesterol from peripheral cells [47], butyrate also plays an important role in the modulation of immune and inflammatory responses and intestinal barrier function [48,49].

Thus, in agreement with our results presented here, other studies that employed metabolomics and NMR identified that pea fiber intake significantly increased the plasma levels of 3-hydroxybutyrate in rats [50]. Therefore, the results suggest that pea fiber can exert antioxidant activity, altering lipid metabolism and decreasing bile acid metabolism [25,50]. On the other hand, Dewulf et al. [51], in a study with obese women who consumed a mix of inulin and oligofructose for three months (16 g/day), reported that there was no significant impact on TC, HDL, LDL and TG compared to the control group, results that do not agree with those of the present study, in which an increase in HDL-C was observed, reinforcing the reduction in BP by cholesterol metabolism.

As limitations of the study, we point out the following aspects: (1) the absence of plasmatic dosages of choline and hydroxybutyrate, (2) that the study population was made up only of women, and (3) that it did not take into account possible genetic backgrounds that may have influenced the response to supplementation. The results found encourage further studies including both genders that seek to elucidate the mechanisms by applying molecular techniques.

## 5. Conclusions

In the present study, there was a significant increase in the concentrations of HDL, choline and hydroxybutyrate and a reduction in BP in hypertensive women with excess weight after intervention with dietary fiber. These significant increases in the different metabolites observed after the intervention were possibly a result of the presence of these metabolites in morbidities such as obesity and hypertension, in addition to those resulting from fiber intake, especially HDL. Therefore, based on the results found in the present study, the intake of mixed dietary fiber should be promoted to combat hypertension. 

## Figures and Tables

**Figure 1 nutrients-13-01437-f001:**
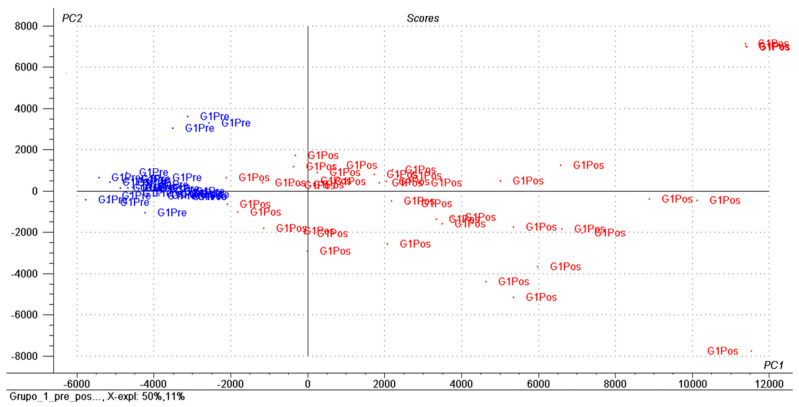
Graphical representation of the principal component analysis (PCA) comparing the metabolomic profiles of hypertensive women with excess weight before and after intervention with mixed dietary fiber (blue-preintervention, red-postintervention).

**Figure 2 nutrients-13-01437-f002:**
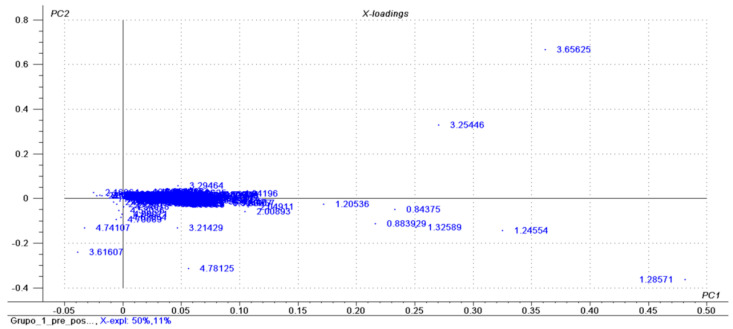
Metabolites identified and highlighted based on the principal component analysis (PCA), presented as a loading graph, comparing the metabolic profiles of hypertensive women with excess weight before and after intervention with mixed dietary fiber.

**Figure 3 nutrients-13-01437-f003:**
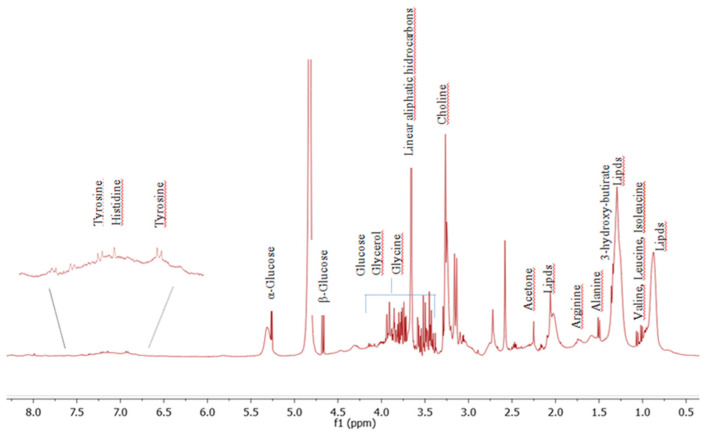
1D Carr Purcell Meiboom Gill (CPMG) 1H NMR spectrum representative of serum metabolites.

**Table 1 nutrients-13-01437-t001:** Clinical and demographic characteristics of the sample of hypertensive women recruited from the Blood Donor Center of João Pessoa, PB, Brazil.

Variable	Mean/SD and %
Age (years)	44.28 ± 9.52
Weight (kg)	88.83 ± 14.25
Height (m)	1.60 ± 0.07
BMI (kg/m²)	34.60 ± 5.90
Overweight	5 (35.7%)
Obese	9 (64.3%)
Use of medication	9 (64.3%)
Education level	
Less than 14 years	7 (38.89)
More than 14 years	7 (38.89)
Household income in US dollars	1243.04 ± 1035.94
Physical activity	5 (27.78%)
Frequency of physical activity/week	1.21 ± 1.89
Tobacco use	1 (0.06%)
Alcohol intake	2 (11.11%)

**Table 2 nutrients-13-01437-t002:** Comparison of the blood pressures and lipid profiles of hypertensive women with excess weight, recruited from the Blood Donor Centre of João Pessoa, PB, Brazil, before and after intervention with mixed dietary fiber.

Variables	Before	After	*p **
Total cholesterol (mg/dL)	180.71 ± 44.78	186.21 ± 39.02	0.5083
TG (mg/dL)	139.93 ± 58.62	134.29 ± 66.86	0.6546
HDL-C	50.5 ± 13.49	58.57 ± 11.76	0.0127 *
LDL-C	105.93 ± 36.89	104.07 ± 28.78	0.8013
VLDL-C	24.36 ± 7.18	23.57 ± 7.99	0.6480
SBP (mmHg)	145.93 ± 19.10	129.64 ± 17.17	0.0013 *
DBP (mmHg)	92.78 ± 12.01	86 ± 10.98	0.0026 *

* Paired *t*-test. VLDL: Very low density lipoprotein. SBP: Systolic blood pressure. DBP: diastolic blood lipoprotein.

**Table 3 nutrients-13-01437-t003:** Comparison between metabolites before and after intervention with mixed dietary fiber in women recruited from the Blood Donor Centre of João Pessoa, PB, Brazil.

Metabolites Signal/NMR PPM ** PPM **	Before	After	*p **
HDL 1.24	492.79 ± 215. 06	1894.39 ± 0.12	0.0010 *
HDL 0.88	747.41 ± 323.87	2245.25 ± 792.05	0.0000 *
HDL 1.32	1048.69 ± 405.98	2669.57 ± 1131.73	0.0000 *
HDL 1.28	1411.59 ± 688.89	4776.78 ± 2082.98	0.0000 *
HDL 4.94	400.48 ± 258.96	959.79 ± 694.40	0.0258 *
Choline 3.25	992.72 ± 285.88	2668.72 ± 1547.87	0.0012 *
Choline 3.65	699.94 ± 877.23	2946.51 ± 2527.32	0.0035 *
Hydroxybutyrate 1.20	191.03 ± 59.24	1537.21 ± 571.08	0.0010 *
Alpha-glucose 4.90	1292.96 ± 949.88	1421.54 ± 958.67	0.9750

* Paired *t*-test or paired Wilcoxon test, based on the distribution of variables. *p* < 0.05. ****** Chemical shift (ppm).

## Data Availability

The data presented in this study are available on request from the corresponding author.

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
