# Peer review of "Choline Metabolites, Hydroxybutyrate and HDL after Dietary Fiber Supplementation in Overweight/Obese Hypertensive Women: A Metabolomic Study"

_nutrients, 2021, doi:10.3390/nu13051437_

Round 1

Reviewer 1 Report

Metabolomic profile of overweight/obese hypertensive women after dietary fibre supplementation

This paper report interesting research on dietary fibre, blood pressure and choline and hydroxybutyrate. However, I have several major concerns about the quality of the paper.

Major issues:

  1. The study do not have a control group and the size of the study is very small only 14 subjects.
  2. In the clinical registration it is specified that the study include 40 normotensive women and 20 hypertensive women. Half the normotensive women should receive control treatment. This is not mentioned in the presented paper. Please describe the selection of data for the presented paper.
  3. Usually it is called metabolomics profile when a number of metabolites are measured and evaluated, however in this paper it is except for HDL only 3 metabolites that are reported.
  4. There is no information at all about the participants diet, not even the background diet which of course is of great importance since the habitual dietary fibre intake could influence the results.
  5. “Therefore, based on the results found in the present study, the intake of mixed dietary fibre should be promoted to combat hypertension.” This is far to strong. A study on 14 subjects without a control group is not evidence. The conclusion could perhaps be that it would be interesting to investigate the findings further.

Suggestions to improve the paper:

  1. Include data on drop-out or side effects are reported.
  2. I would suggest to change the title because this is not a metabolomic profile. I would suggest the authors to focus this paper on choline and hydroxybutyrate. The changes in these are interesting findings.
  3. If dietary data exists include this in the paper.
  4. The authors claim that there is a lack of studies on dietary fibre and metabolomics profile: ” In the literature consulted, there were no studies related to the metabolomic profile of overweight and obese hypertensive women who consumed mixed dietary fibre.” However, the study presented are not a study on metabolomics profile. Its possible to call it “certain metabolites”. Maybe the NMR-method is the limitation, because we at least identify 60 metabolites with our NMR-method or is only a few metabolites selected and presented?
  5. Language editing. Many sentences are long and very hard to read. Please divide into several sentences. The last sentence in abstract is very hard to understand.
  6. “Considering the persistent increase in the prevalence of hypertension, the need for innovative prevention and control strategies has become important, using new technologies to assist in obtaining more robust results.” This is an example. The last part doesn´t make any sense. Is this what the authors mean?

Considering the persistent increase in the prevalence of hypertension, the need for innovative prevention and control strategies has become important. Here, new technologies can be used to assist in obtaining more robust results.

  1. References placed in the middle of sentences.
  2. The decrease in BP values are due to soluble dietary fibre as well as to an increase in intestinal viscosity, delaying nutrient absorption and inhibiting the absorption of cholesterol and bile acids [18,19]. This is not connected to why this would decrease BP? Please explain this.
  3. In this study, only women diagnosed with primary hypertension with a body mass index (BMI) of 25 - 35 m²/kg and BP above 130/80 mmHg were included in the study. Remove the last part.
  4. The women who agreed to participate in the study kept their eating… What do you mean by eating? Diet?
  5. BLOOD PRESSURE ASSESSMENT- were the patients resting before? Lying down or sitting. This is the main outcome so describe carefully.
  6. With these procedures, it was possible to differentiate the phases before and after intervention with mixed dietary fibre. Should not be a part of the method.
  7. Figure 1 and 2 (that is called graph) has a very low quality. The text is to small and not readable.
  8. “Therefore, the high choline values found in the present study, after the intervention with fibre, were probably increased due to the existence of high amounts of this metabolite in hypertension and obesity; however, some studies with the described morbidities did not find an increase in choline.” I do not understand this. The blood pressure decreased and the choline increased.
  9. Why is the authors discussing metabolites in general in subjects that are obese or have hypertension? Since the paper doesn´t include a normal weight control group this is not the scope of the study. The discussion is not focused on the study results and must be rewritten.

Author Response

Response to Reviewer 1 Comments

Point 1: I. The study do not have a control group and the size of the study is very small only 14 subjects.

 Response 1: At the moment, the present study has no control group, because since the theme is unprecedented in the international literature, the team of researchers chose for the in-depth study on the topic related to the metabolic profile of overweight / obese hypertensive women after intervention with mixed dietary fibre.

The size of the study group is in accordance with the sample calculation by group, since in the larger research project, there were four groups that were divided into normotensive and hypertensive individuals who used fibre and placebo, which was based on the effect of the fibre on the pressure values in a pilot study. Reliability of 95% (error α = 1.96) and margin error of 5% and β error with power of 80% (0.84) were adopted and the standard deviation of this distribution was estimated, considering the fibre effect difference of 12mmHg. Thus, N of at least 11 women per group was obtained.

Point 2: In the clinical registration it is specified that the study include 40 normotensive women and 20 hypertensive women. Half the normotensive women should receive control treatment. This is not mentioned in the presented paper. Please describe the selection of data for the presented paper.

Response 2: This study is part of a larger project entitled: "Effect of dietary fibre supplementation on blood pressure, metabolites and cardiovascular risk score in adult overweight women with c677t polymorphism in the mthfr gene".

There was a mistake in the Clinical Trials record, since there were four groups of 20 women divided into normotensive and hypertensive using placebo and fibre, totalling 80 women. Therefore, 60 women were included instead of 80 women.

The present study worked with the group of hypertensive women who consumed dietary fibres, which at the beginning were 20 women. Thus, 05 women did not meet the eligibility criteria and only one participant withdrew due to the side effects of dietary fibre such as colic and diarrhoea.

Point 3: Usually it is called metabolomics profile when a number of metabolites are measured and evaluated, however in this paper it is except for HDL only 3 metabolites that are reported.

 Response 3: Suggestion accepted also to change the title: “Choline metabolites, hydroxybutyrate and HDL after dietary fibre supplementation in overweight / obese hypertensive women: a metabolomic study”

Point 4: There is no information at all about the participants diet, not even the background diet which of course is of great importance since the habitual dietary fibre intake could influence the results

Response 4: Yes. There are data on participants' diet. The following information was added to the methodology and results sections.

In the methodology section in the last paragraph of point 2.4. NUTRITIONAL EVALUATION precisely from line 6 and in the results section in the sixth line at the bottom of table 1.

The usual intake of total calories, carbohydrates, proteins, lipids and fibres was 2390.91 ± 1449.42 kcal, 267.82 ± 103.26 g, 68.29 ± 25.69g, 61.06, ± 30.90 g and 15.57 ± 5.59 g, respectively after analysis of two recalls per individual applied on the first day of the study and the other after 8 weeks using the Virtual Nutri Plus software and later the intra and interpersonal adjustment using the MSM software.

 Point 5: “Therefore, based on the results found in the present study, the intake of mixed dietary fibre should be promoted to combat hypertension.” This is far to strong. A study on 14 subjects without a control group is not evidence. The conclusion could perhaps be that it would be interesting to investigate the findings further.

Response 5: Suggestion accepted: "Therefore, based on results found in the present study, it is suggested to further investigate the results of the intake of mixed dietary fibres in the control of hypertension."

Suggestions to improve the paper:

  1. Include data on drop-out or side effects are reported.

Suggestion accepted. Only one participant withdrew due to the side effects of dietary fibre such as colic and diarrhoea. The following information was added to the methodology : 2.1. Study design and sample characteristics on line 8.

2.I would suggest to change the title because this is not a metabolomic profile. I would suggest the authors to focus this paper on choline and hydroxybutyrate. The changes in these are interesting findings.

Change of title accepted. "Choline metabolites, hydroxybutyrate and HDL after dietary fibre supplementation in overweight / obese hypertensive women: a metabolomic study"

3.If dietary data exists include this in the paper.

           Yes. Suggestion accepted and included in the article. In the methodology section in the last paragraph of point 2.4. NUTRITIONAL EVALUATION

4.The authors claim that there is a lack of studies on dietary fibre and metabolomics profile: ” In the literature consulted, there were no studies related to the metabolomic profile of overweight and obese hypertensive women who consumed mixed dietary fibre.” However, the study presented are not a study on metabolomics profile. Its possible to call it “certain metabolites”. Maybe the NMR-method is the limitation, because we at least identify 60 metabolites with our NMR-method or is only a few metabolites selected and presented?

The metabolites that most stood out statistically, that is, those that had the highest peak intensity were presented in the study, and the main ones were HDL, choline, alpha glucose and hydroxybutyrate. However, amino acids were identified according to the spectrum in figure 1.

5.Language editing. Many sentences are long and very hard to read. Please divide into several sentences. The last sentence in abstract is very hard to understand.

6.“Considering the persistent increase in the prevalence of hypertension, the need for innovative prevention and control strategies has become important, using new technologies to assist in obtaining more robust results.” This is an example. The last part doesn´t make any sense. Is this what the authors mean?

Considering the persistent increase in the prevalence of hypertension, the need for innovative prevention and control strategies has become important. Here, new technologies can be used to assist in obtaining more robust results.

7.References placed in the middle of sentences.        

  Yes. Suggestion accepted

8.The decrease in BP values are due to soluble dietary fibre as well as to an increase in intestinal viscosity, delaying nutrient absorption and inhibiting the absorption of cholesterol and bile acids [18,19]. This is not connected to why this would decrease BP? Please explain this.

Yes. The reduction in BP as a result of fibre consumption has as main justification the involvement with cholesterol metabolism. Thus, this paragraph justifies the mechanism by which dietary fibre lowers blood pressure. Fibre is involved in the metabolism of bile acids, promoting absorption of lipids and acting in the cholesterol metabolism. Thus, fibre appears to decrease the lipid profile by increasing intestinal viscosity, reducing the absorption of bile acids and promoting cholesterol catabolism, interfering in blood pressure values.

9.In this study, only women diagnosed with primary hypertension with a body mass index (BMI) of 25 - 35 m²/kg and BP above 130/80 mmHg were included in the study. Remove the last part.

            Changed in the article in point 2.1. Study design and sample characteristics on line 18

10.The women who agreed to participate in the study kept their eating… What do you mean by eating? Diet?

Selected women who agreed to participate in the study did not change their diet, that is, they kept their eating habits during intervention.

11.BLOOD PRESSURE ASSESSMENT- were the patients resting before? Lying down or sitting. This is the main outcome so describe carefully.

            Women were resting (at least 05 minutes) and in the sitting position with feet on the  ground and arms supported on the table.The following information was added to the methodology :2.3. BLOOD PRESSURE   ASSESSMENT on line 5.

12.With these procedures, it was possible to differentiate the phases before and after intervention with mixed dietary fibre. Should not be a part of the method.

Suggestion accepted. Excluded from the method section.

13.Figure 1 and 2 (that is called graph) has a very low quality. The text is to small and not readable.

Suggestion accepted. Chart modified with better quality

14.Thefore, the high choline values found in the present study, after the intervention with fibre, were probably increased due to the existence of high amounts of this metabolite in hypertension and obesity; however, some studies with the described morbidities did not find an increase in choline.” I do not understand this. The blood pressure decreased and the choline increased.

The text was excluded from the article.

15.Why is the authors discussing metabolites in general in subjects that are obese or have hypertension? Since the paper doesn´t include a normal weight control group this is not the scope of the study. The discussion is not focused on the study results and must be rewritten.

We would like to clarify that our sample was composed of hypertensive and overweight women. The association between obesity and hypertension in the same individual is very common in population-based studies, and the results of studies could better propose adequate treatment for this population.

Reviewer 2 Report

Fechine et al. evaluated the metabolomic profile of 14 overweight and obese hypertensive women before and after an intervention with mixed fiber. 8 weeks later, peak intensity of HDL-C, choline, and hydroxybutyrate significantly increased significantly, and systolic and diastolic blood pressure decreased.

In their discussion, they stated that betaine is derived from choline and this metabolite is also positively related to obesity [27,28], lipid metabolism [29], and hypertension [30], therefore, their study indirectly supports the increase in choline seen in the their study. However, there is no mention of body weight changes in this study, and the lack of changes in total cholesterol, TG, and alpha-glucose make it unlikely that there was an excess energy intake. This does not seem to support an increase in choline. Furthermore, cholinergic metabolites are found to change in obesity, which indirectly supports the significant increase in choline in this study, but since it is unlikely that obesity progressed in 8 weeks (rather, isn't it in the direction of improvement?), the choline increase may be another reason.

A study conducted by Wiklund [33] found that branched-chain amino acids, aromatic amino acids, several fatty acids and phospholipids were associated with metabolic syndrome, and this result [33] is stated to partially support the results of this study, but there is no mention of such a result in the text, it is unclear whether they are supported.

In the discussion, many papers are cited to explain the variations and relationships among various metabolites, but the paper would be easier for readers to understand if they were presented as figures. From the current description alone, it is very difficult to understand the relationship between the present results and previous papers, and in the end, it is unclear what the present study has revealed.

Author Response

Response to Reviewer 2 Comments

Point 1: In their discussion, they stated that betaine is derived from choline and this metabolite is also positively related to obesity [27,28], lipid metabolism [29], and hypertension [30], therefore, their study indirectly supports the increase in choline seen in the their study. However, there is no mention of body weight changes in this study, and the lack of changes in total cholesterol, TG, and alpha-glucose make it unlikely that there was an excess energy intake. This does not seem to support an increase in choline. Furthermore, cholinergic metabolites are found to change in obesity, which indirectly supports the significant increase in choline in this study, but since it is unlikely that obesity progressed in 8 weeks (rather, isn't it in the direction of improvement?), the choline increase may be another reason.

Response 1: I agree with your observation. In the present study, there was no statistically significant weight change, that is, increase in the weight of participants after intervention with dietary fibre. Therefore, the increase in choline contents must have probably occurred due to other reasons that should be investigated in the future.

Thus, so far, there is no justification in literature for the increase in choline levels from supplementation with dietary fibres, making further studies necessary.

Point 2: A study conducted by Wiklund [33] found that branched-chain amino acids, aromatic amino acids, several fatty acids and phospholipids were associated with metabolic syndrome, and this result [33] is stated to partially support the results of this study, but there is no mention of such a result in the text, it is unclear whether they are supported.

Response 2: I justified the author's statement in the seventh paragraph of the discussion section.The results [33] partially corroborate those of the present study, which also identified the hydroxybutyrate metabolite and an increase in an important phospholipid, in this case phosphatidylcholine, which is rich in choline, as well as branched-chain amino acids (valine, leucine and isoleucine), histidine, tyrosine, arginine, and alanine, although they were not significant different before and after the intervention with dietary fibre.

Point 3: In the discussion, many papers are cited to explain the variations and relationships among various metabolites, but the paper would be easier for readers to understand if they were presented as figures. From the current description alone, it is very difficult to understand the relationship between the present results and previous papers, and in the end, it is unclear what the present study has revealed.

Response 3: Please, I would like an explanation about “what does it means to be presented as numbers” in the discussion section for the better understanding of readers.

Round 2

Reviewer 1 Report

The authors have added a sentence in Spanish about the diet. Please write in english. Also the authors have added data but there is not much description on how the data was collected. Recalls- is that 24h recalls, who did this- a dietitian? Also please separate data from baseline and at eight weeks to show differences between baseline and end and test if there are changes. An addition of fibre may have had effects on the overall diet.

Language editing. Many sentences are long and very hard to read. I asked the authors to make the paper easier to read and they have changed only the sentences I used as an example.

Author Response

Response to Reviewer 1 Comments

Point 1: The authors have added a sentence in Spanish about the diet. Please write in english.

We appreciated the referee's recommendation and corrected the sentence that was written in another language.

Point 2.Also the authors have added data but there is not much description on how the data was collected. Recalls- is that 24h recalls, who did this- a dietitian? Also please separate data from baseline and at eight weeks to show differences between baseline and end and test if there are changes. An addition of fibre may have had effects on the overall diet.

The requested information was included in the methodology and results.

The usual intake of total calories, carbohydrates, proteins, lipids and fibres were analysed using two recalls (24-hour food recall), applied by by professional nutritionists on the first day of the study and after 8 weeks. Food consumption was analyzed using the Virtual Nutri Plus software and subsequently the intra and interpersonal adjustment using the MSM software (Multiple Sources method).

The usual intake of total calories. carbohydrates, proteins. lipids and fibers, was 2390.91 ± 1449.42 kcal, 267.82 ± 103.26 g, 68.29 ± 25.69g, 61.06, ± 30.90 g and 15.57 ± 5.59 g, respectively after analysis of two 24-hour food recall, applied on the first day of the study and at 8 weeks. The usual intake of dietary fibers in the diet, before the intervention and after the intervention, was 15, 57 + _ 5.59 g and that added to 12 g of fibers from the intervention with mixed dietary fibers, totaled 27 , 57 g of mixed fiber consumption during the 8 weeks of the study

Point 3.Language editing. Many sentences are long and very hard to read. I asked the authors to make the paper easier to read and they have changed only the sentences I used as an example.

We appreciate the opportunity to improve article writing. The text has been completely revised and the long sentences have been replaced by short, objective sentences. All changes are highlighted in yellow.
